# Chloroaluminate Ionic Liquid Immobilized on Magnetic Nanoparticles as a Heterogeneous Lewis Acidic Catalyst for the Friedel–Crafts Sulfonylation of Aromatic Compounds

**DOI:** 10.3390/molecules27051644

**Published:** 2022-03-02

**Authors:** Ngoc-Lan Thi Nguyen, Quoc-Anh Nguyen, Tien Khoa Le, Thi Xuan Thi Luu, Kim-Ngan Thi Tran, Phuoc-Bao Pham

**Affiliations:** 1Faculty of Chemistry, University of Science, 227 Nguyen Van Cu, Dist. 5, Ho Chi Minh 700000, Vietnam; lanbmt07@gmail.com (N.-L.T.N.); nquocanh1996@gmail.com (Q.-A.N.); ltkhoa@hcmus.edu.vn (T.K.L.); thikimngan.tran@gmail.com (K.-N.T.T.); pphuocbao@gmail.com (P.-B.P.); 2Faculty of Chemistry, Vietnam National University of Hochiminh City, Ho Chi Minh 700000, Vietnam

**Keywords:** sulfonylation, magnetic nanoparticles, sulfones, ionic liquids, sulfonic anhydride

## Abstract

Chloroaluminate ionic liquid bound on magnetic nanoparticles (Fe_3_O_4_@O_2_Si[PrMIM]Cl·AlCl_3_) was prepared and used as a heterogenous Lewis acidic catalyst for the Friedel–Crafts sulfonylation of aromatic compounds with sulfonyl chlorides or *p*-toluenesulfonic anhydride. The catalyst’s stability, efficiency, easy recovery, and high recyclability without considerable loss of catalytic capability after four recycles were evidence of its advantages. Furthermore, the stoichiometry, wide substrate scope, short reaction time, high yield of sulfones, and solvent-free reaction condition also made this procedure practical, ecofriendly, and economical.

## 1. Introduction

Sulfones, one of the most common organosulfur compounds, have tremendous applications in chemical processes [1,2], medicinal chemistry, and drug syntheses owing to their various biological activities; for instance, anti-inflammatory [3,4], anti-HIV [5], antimalarial [6,7], anticancer [8], and antimicrobial [9,10], and as a cysteine protease inhibitor [11].

Widespread synthetic routes of sulfones via the oxidation of the corresponding sulfides or sulfoxides [12,13,14], the sulfonylation of chloropyridine derivatives by sulfinate salts [15], the arylation of sulfinate salts by diaryliodonium salts [16], the formation of a C–S bond via the reaction of various silyl triflate and arenesulfinate salts [17], the addition to alkynes by sulfinate salts [18], the oxidative cyclization of phenyl propiolates with sulfinic acids initiated by visible light [19], the decarboxylative C–S cross-coupling of cinnamic acid with benzenesulfinate salts promoted by iodine [20], and the Friedel–Crafts sulfonylation have been developed. Among numerous approaches to sulfone preparations, aryl sulfones have been synthesized, preferably via the Friedel–Crafts sulfonylation reactions between activated arenes and sulfonylating reagents in the presence of catalysts; e.g., Lewis acidic salts [21,22,23,24,25,26], Zn [27], In/dioxane [28], MoO_2_Cl_2_ [29], metal triflate [30,31], Fe(OH)_3_ [32], nafion-H [33], Fe(III)-exchanged montmorillonite [34], Ps-AlCl_3_ and SiO_2_-AlCl_3_ [35,36], Lewis acidic salt-based ionic liquids [37,38], and P_2_O_5_ supported on Al_2_O_3_ [39] or SiO_2_ [40].

Within the tendency of scientific and technological improvement, environmental assessment has been mainly paid attention. In recent decades, ionic liquids (ILs), as well as functionalized ionic liquids, play important roles as solvents and homogeneous catalysts in several organic synthesis processes, owing to their low vapor pressure, thermal stability, high ability to dissolve many inorganic and organic compounds [41]. Homogeneous catalysts are always dissolved easily in various organic solvents or reaction media, therefore it is difficult to recover and recycle catalysts used. Contrarily, heterogeneous catalysts could be recovered and recycled conveniently and efficiently, although their dispersion in reaction media have not been carried out well. To overcome these problems in the dispersion, recovery, and recycling of catalysts, ionic liquids have been immobilized onto solid materials such as organic polymers [42,43,44], inorganic supports (e.g., silica, alumina) [45,46,47,48], and magnetic nanoparticles (MNPs) [49,50,51,52,53,54,55]. The improved catalysts have possessed the combined properties of homogeneous and heterogeneous catalysts, consisting of a larger surface area and catalyst-loading capacity, better dispersity in reaction media, and simple separation. In general, MNPs are selected as excellent solid supports for ILs, owing to a convenient removal of the catalyst by using an external magnet without filtration or centrifugation [56].

Using the advantages of magnetic nanoparticles in catalysis, in this work, we developed a magnetic nanoparticle–Fe_3_O_4_ linked acidic ionic liquid as a green and efficient catalyst to be used for the Friedel–Crafts sulfonylation of activated arenes or polyarenes with sulfonyl chloride or sulfonic ahydride (Figure 1). Magnetic nanoparticle–Fe_3_O_4_ linked acidic ionic liquids have been used for several transformations, such as three-component reactions of benzaldehyde derivatives, urea/thiourea, and acetoacetate [50]; benzaldehyde derivatives, *β*-naphthol, and 1,3-cyclohexandione derivatives [57]; and benzaldehyde derivatives, aniline derivatives, and 2-mercaptoethanoic acid [58].

## 2. Results and Discussion

At the beginning of this work, based on the disadvantages of the recovery and recycling of the chloroaluminate ionic liquid used for the Friedel–Crafts sulfonylation between toluene and benzenesulfonyl chloride [59], several magnetic nanoparticles bound by a Lewis/Brønsted acidic ionic liquid, such as Fe_3_O_4_@O_2_Si[PrMIM]HSO_4_, MgFe_2_O_4_@O_2_Si[PrMIM]Cl·AlCl_3_, and Fe_3_O_4_@O_2_Si[PrMIM]Cl·AlCl_3_, were developed and evaluated under a solvent-free sulfonylation reaction (entries 1–3, Table 1). Consequently, Fe_3_O_4_@O_2_Si[PrMIM]Cl·AlCl_3_ was selected as the best acidic catalyst among the heterogeneous catalysts used, owing to its efficiency (Table 1).

### 2.1. Catalyst Characterization

Magnetic fine particles were continuously prepared by coprecipitation of iron(II) and iron(III) salts at 80 °C. The precipitated fine particles were characterized by XRD for the structural determination (Figure 1), and by FT-IR spectra (Figure 2a) and SEM for the crystallite size (Figure 3a). The XRD pattern of Fe_3_O_4_ showed that five diffraction peaks appeared at around 30.20°, 35.56°, 43.14°, 57.06°, and 62.59°, which corresponded to the crystallographic planes ((220), (311), (400), (511), and (440) lines, respectively) of the magnetite Fe_3_O_4_ phase [60]. In addition, the SEM micrograph of the Fe_3_O_4_ also displayed that cubic-shaped particles in agglomerated states reached a nanoparticle diameter of approximately 20.0 nm (Figure 3a). Subsequently, the heterogeneous catalyst, Fe_3_O_4_@O_2_Si[PrMIM]Cl·AlCl_3_, was prepared from magnetic nanoparticles, 3-methyl-1-(3-trimethoxysilylpropyl)-1H-imidazol-3-ium chloride, and aluminum chloride as described in Figure 2, and then characterized by XRD (Figure 1), FT-IR (Figure 2c), SEM and TEM (Figure 3b,c), EDX (Figure 4), TGA (Figure 5), VSM (Figure 6), BET, and ICP-MS.

In the XRD pattern of the Fe_3_O_4_@O_2_Si[PrMIM]Cl·AlCl_3_ sample, these characteristic peaks were still present, but their intensities were dramatically decreased. The presence of ionic liquid in this sample was able to affect to the crystallinity of the magnetite phase (Figure 1).

The influence of the ionic liquid on the surface of the Fe_3_O_4_ was also investigated via FT-IR spectra (Figure 2). In the FT-IR spectrum of Fe_3_O_4_, three peaks were clearly detected at 3425, 1625, and 585 cm^−1^, which were respectively attributed to O–H stretching, O–H bending, and Fe–O stretching vibrations. When the Fe_3_O_4_ particles were combined with the ionic liquid, new peaks were observed at 2950 and 1082 cm^−1^, in which the signal at 2950 cm^−1^ was obviously assigned to the aliphatic C–H stretching vibration of the propyl group, and the latter signal at 1082 cm^−1^ belonged to the Si–O stretching vibration. This proved that the immobilization of the ionic liquid on the Fe_3_O_4_ surface occurred successfully.

The surface morphology of the Fe_3_O_4_@O_2_Si[PrMIM]Cl·AlCl_3_ was also compared with that of the Fe_3_O_4_ by scanning electron microscopy (SEM). As shown in Figure 3a, the Fe_3_O_4_ sample consisted of agglomerated particles with sizes varying from 20 to 40 nm. Interestingly, the SEM and TEM images of Fe_3_O_4_@O_2_Si[PrMIM]Cl·AlCl_3_ (Figure 3b,c) showed the presence of a liquid layer covering the surface of the magnetic particles. The size distribution of the magnetic nanoparticles modified by chloroaluminate ionic liquid varied in the range of 6 nm to 14 nm (Figure 3d). The aggregation of nanoparticles prevented by the presence of chloroaluminate ionic liquid immobilized on magnetic nanoparticles was the reason for the size reduction of the Fe_3_O_4_ particles.

The elemental composition determined by energy dispersive X-ray spectroscopy (EDX) illustrated that the catalyst contained carbon (C), chlorine (Cl), aluminum (Al), oxygen (O), and silicon (Si), which were the characteristic elements of the chloroaluminate ionic liquid (Figure 4). Moreover, according to the results of an inductively coupled plasma mass spectrometry (ICP-MS) analysis and nitrogen absorption experiments, the aluminum content and the BET specific surface area of the Fe_3_O_4_@O_2_Si[PrMIM]Cl·AlCl_3_ were found to be 1.12 mmol g^−1^ and 74 m^2^ g^−1^, respectively.

In order to investigate the thermal stability of our catalyst, a thermogravimetric diagram of the Fe_3_O_4_@O_2_Si[PrMIM]Cl·AlCl_3_ was recorded by heating the sample up to 600 °C (Figure 5). The diagram illustrated a slight weight loss of 7% below 300 °C, owing to the evaporation of adsorbed water. From 320–460 °C, a sharp decrease in weight observed (approximately 15%) was caused by the decomposition of imidazole moieties [61]. These results did not only confirm the fact that organic parts had been successful grafted on magnetic nanoparticles, but also determined the thermal stability of our catalyst up to 300 °C.

The magnetic parameters of the Fe_3_O_4_ and ionic liquid-coated Fe_3_O_4_ were identified using a vibrating sample magnetometer (VSM) at room temperature (Figure 6). The absence of a hysteresis loop in the obtained VSM curves substantiated our catalyst as a superparamagnetic material. Due to the grafting processes, the saturation magnetization value (*M_s_*) of the Fe_3_O_4_@O_2_Si[PrMIM]Cl·AlCl_3_ (32.64 emu/g) was lower than that of the Fe_3_O_4_ (34.99 emu/g); however, the *M_s_* value of the Fe_3_O_4_@O_2_Si[PrMIM]Cl·AlCl_3_ was still high enough for the separation of the catalyst out of the reaction mixture by using an external magnet.

### 2.2. Friedel–Crafts Sulfonylation

In the next experiments, the amount of completed catalyst, Fe_3_O_4_@O_2_Si[PrMIM]Cl·AlCl_3_, was investigated in detail to improve the yield of sulfone (entries 3–5, Table 1). Molar ratios of toluene and benzenesulfonyl chloride varying from 1.0:1.0 up to 1.5:1.0 (mmol/mmol) in 0.1 mmol increments for toluene, as well as reaction temperatures in the range of 80–110 °C in 10 °C increments were used. Finally, the appropriate amount of toluene (1.4 mmol), benzenesulfonyl chloride (1.0 mmol), and Fe_3_O_4_@O_2_Si[PrMIM]Cl·AlCl_3_ (0.2 g) were selected and used in solvent-free sulfonylation for four hours at 110 °C (entry 1, Table 2). Further experiments on the nature of alkanesulfonyl/arenesulfonyl chloride were investigated (entries 1–12, Table 2). The results of eight experiments between four arenesulfonyl chlorides and toluene, as well as anisole, displayed that the electron-withdrawing substituents on the aromatic ring of arenesulfonyl chloride caused lower yields of sulfone than electron-donating groups. In addition, three alkanesulfonyl chloride reactions with anisole were also performed; however, the amount of product mixture obtained was much lower than in the case of arenesulfonyl chloride with anisole. In these cases, the sulfonylium cation in transition state stabilized by the aromatic ring better than the aliphatic carbon chain was the main reason for the lower yield of the newborn sulfone obtained from the reactions of three alkanesulfonyl chlorides with anisole (entries 10–12, Table 2). Similarly, in the next series of experiments, *p*-toluenesulfonyl chloride was chosen as the sulfonylating reagent to investigate the influences of the structure of aromatic compounds on the yields of sulfones (entries 13–18, Table 2). Consequently, the Friedel–Crafts sulfonylation preferred the activated aromatic rings to afford the corresponding sulfones in good yields—the more electron-donating substituents on the aromatic ring, the more the yields of sulfones. Therefore, 1-chloro-4-tosylbenzene was formed at a low yield for a longer reaction time (entry 13, Table 2) owing to the chlorine substituent, a deactivated group linked to the benzene ring. With the mild and efficient catalyst, Fe_3_O_4_@O_2_Si[PrMIM]Cl·AlCl_3_, demethylation of the methoxy-substituted group was not detected in most experiments by gas chromatography–mass spectrometry analyses (GC/MS), as well as thin-layer chromatography (TLC), in comparison with strong Lewis acidic as the aluminum chloride. Selectively, sulfonyl groups were located at the *para* position with the available substituents on aromatic rings better than those at the *ortho* position in the Friedel–Crafts sulfonylation of monosubstituted benzene rings. In order to enlarge the scope of substrates used for this process, a polycyclic benzenoid hydrocarbon; e.g., naphthalene or dibenzothiophene, were also selected as model substrates to react with the excess amount of arenesulfonyl chlorides as the reactant and the solvent so that average to fair yields were obtained (entries 20–21, Table 2).

In another experiment, the sulfonylating reagent arenesulfonyl chloride was replaced with sulfonic anhydride to produce diaryl sulfones in the Friedel–Crafts sulfonylation of activated aromatic compounds (Table 3). Although the yields of sulfones obtained by using sulfonic anhydride were a little bit lower than those by using sulfonyl chloride, *p*-toluenesulfonic anhydride showed its capability as a moderately efficient, mild, and alternative reagent for the Friedel–Crafts sulfonylation. Finally, the above results substantiated our choice of Fe_3_O_4_@O_2_Si[PrMIM]Cl·AlCl_3_ as the most efficient catalyst for both sulfonylating reagents, sulfonyl chloride and sulfonic anhydride. It not only caused the reaction to occur in mild and solvent-free media, but also improved the isolation of sulfones, as well as the separation of catalyst (Table 3).

With the advantages of Fe_3_O_4_@O_2_Si[PrMIM]Cl·AlCl_3_ in the enhancement of reactivity and recovery of catalyst, the reusability of Fe_3_O_4_@O_2_Si[PrMIM]Cl·AlCl_3_ was examined. Fe_3_O_4_@O_2_Si[PrMIM]Cl·AlCl_3_ was collected after separation with an external magnet, washed alternately with ethanol (2 × 5 mL) and acetone (2 × 5 mL), and dried in a desiccator overnight. The recovered catalyst was obtained at a yield of 93% and analyzed by FT-IR. The FT-IR analysis demonstrated that the functional groups of the recovered catalyst in the fourth recycle were compatible with those of the fresh Fe_3_O_4_@O_2_Si[PrMIM]Cl·AlCl_3_ (Figure 7). Simultaneously, the recycled Fe_3_O_4_@O_2_Si[PrMIM]Cl·AlCl_3_ was used for the sulfonylation of toluene with benzenesulfonyl chloride at 110 °C for four hours, as in the optimal experiment mentioned in entry 1 of Table 2. The catalytic efficiency of the Fe_3_O_4_@O_2_Si[PrMIM]Cl·AlCl_3_ did not change considerably, even after four cycles of catalyst recovery and reuse (Figure 8).

The introduced protocol of the sulfone synthesis from the Friedel–Crafts sulfonylation promoted by Fe_3_O_4_@O_2_Si[PrMIM]Cl·AlCl_3_ offered several advantages in terms of a lower amount of aromatic compounds used; a green, efficient and economic catalyst; and a high product selectivity and yield under the solvent-free reaction condition compared with the results in the previous literature reported on Friedel–Crafts sulfonylation with different catalysts (Table 4).

## 3. Materials and Methods

Sulfonyl chlorides (benzenesulfonyl chloride, 4-methylbenzenesulfonyl chloride, ethanesulfonyl chloride, isobutanesulfonyl chloride, …), anhydrous aluminum chloride, arenes (anisole, 1,3-dimethoxybenzene, naphthalene, chlorobenzene, …), (3-chloropropyl)trimethoxysilane, and 1-methylimidazole were from Sigma-Aldrich (Darmstadt, Germany), and the *p*-toluenesulfonic anhydride and isomer of xylene were from Acros. All commercially available chemicals were analyzed for authenticity and purity by GC/MS before being used. X-ray diffraction patterns were measured on a Brüker D8 Advance diffractometer. Fourier-transform infrared (FT-IR) spectra were recorded on a Brüker E400 spectrometer in the range of 4000–500 cm^−1^. Thermal gravimetric analysis (TGA) was performed using a TA Instruments Q-500 thermal gravimetric analyzer. Magnetic properties were measured using an ID-EV 11 vibrating sample magnetometer (VSM). Size and structure of materials were obtained using a Hitachi S-4800 scanning electron microscope (SEM) and JOEL JEM1010 transmission electron microscope (TEM). The composition of the catalyst was analyzed by energy-dispersive X-ray spectroscopy (EDX) on a Shimadzu EDX-8000. The specific surface area was determined using the Brunauer–Emmett–Teller (BET) technique with a Quantachrome NOVA 2200e analyzer (Boynton Beach, FL, USA). Inductively coupled plasma mass spectroscopy (ICP-MS) data were recorded on an Agilent 7700s instrument. NMR spectra were recorded on a Brüker AVANCE 500 or Brüker AVANCE NEO 400 at 500 or 400 MHz for ^1^H-NMR and 125 or 100 MHz for ^13^C-NMR. Gas chromatography analyses were performed on an Agilent 6890, with a flame ionization detector equipped with a J and W DB-5MS capillary column (30 m, 0.25 mm i.d., 0.25 µm film thickness). Gas chromatography–mass spectrometry (GC-MS) measurements were carried out on an Agilent GC System 7890 equipped with a mass selective detector (Agilent 5973N) and a capillary DB-5MS column (30 m × 250 µm × 0.25 µm). High-resolution mass spectrometry (HRMS) was recorded on an Agilent 1200 series high-performance liquid chromatograph with a Bruker micrOTOF-QII EIS mass spectrometer detector.

### 3.1. General Procedure for Preparation of Heterogeneous Catalyst Fe_3_O_4_@O_2_Si[PrMIM]Cl·AlCl_3_

#### 3.1.1. The Preparation of MNPs via the Modified Chemical Coprecipitation Method 

Typically, 100 mL of FeSO_4_·7H_2_O (6.0 mmol, 1.668 g) and Fe(NO_3_)_3_·9H_2_O (12.0 mmol, 4.848 g) dissolved completely in 100 mL distilled water was dropped slowly into a 500 mL beaker containing 200 mL of 0.25 M NaOH solution within 1 h at 80 °C under vigorous mechanical stirring at 500 rpm. The black precipitate was washed with distilled water (2 × 100 mL) until reaching pH 7 and dried at 150 °C for 4 h. The crude iron oxide particles were ground with a porcelain mortar to obtain the fine magnetic nanoparticles (MNPs) [56].

#### 3.1.2. The Preparation of 3-Methyl-1-(3-trimethoxysilylpropyl)-1H-imidazole-3-ium Chloride

A mixture of (3-chloropropyl)trimethoxysilane (20.0 mmol, 3.974 g) and 1-methylimidazole (20.0 mmol, 1.642 g) in a round-bottom 25 mL flask was stirred at 80 °C for 72 h. After reaction completion, the mixture of products was washed with diethyl ether (3 × 5 mL). Subsequently, the pure ionic liquid with light yellow, 3-methyl-1-(3-trimethoxysilylpropyl)-1H-imidazole-3-ium chloride obtained after the solvent removal under vacuum pressure was identified by ^1^H and ^13^C NMR spectroscopy. These spectra were compatible with the previous literature [56].

#### 3.1.3. Methyl-1-(3-trimethoxysilylpropyl)-1H-imidazole-3-ium Chloride

Methyl-1-(3-trimethoxysilylpropyl)-1H-imidazole-3-ium chloride, light yellow liquid. ^1^H NMR (500 MHz, CDCl_3_): δ (ppm) 10.56 (brs, 1H), 7.46 (s, 1H), 7.32 (s, 1H), 4.29 (t, *J* = 7.5 Hz, 2H), 4.09 (s, 3H), 3.54 (s, 9H), 1.98 (p, *J* = 7.5 Hz, 2H), 0.63–0.59 (m, 2H). ^13^C NMR (125 MHz, CDCl_3_): δ (ppm) 138.5, 123.3, 121.8, 51.9, 50.8, 36.8, 24.2, 6.1.

#### 3.1.4. The Preparation of Fe_3_O_4_@O_2_Si[PrMIM]Cl

Fe_3_O_4_ nanoparticles (1.0 mmol, 0.232 g), 3-methyl-1-(3-trimethoxysilylpropyl)-1H-imidazole-3-ium chloride (2.0 mmol, 0.562 g), absolute ethanol (5.0 mL), and 28% ammonia solution (0.2 mL) were added into a round-bottom 25 mL flask and stirred at room temperature for 24 h. After reaction completion, Fe_3_O_4_@O_2_Si[PrMIM]Cl, a dark-brown solid, was washed with ethanol (2 × 5 mL) and collected with an external magnet and then dried under vacuum.

#### 3.1.5. The Preparation of Fe_3_O_4_@O_2_Si[PrMIM]Cl·AlCl_3_

Anhydrous aluminum chloride, AlCl_3_ (4.0 mmol, 0.533 g), was added slowly into a 25 mL round-bottom flask containing Fe_3_O_4_@O_2_Si[PrMIM]Cl dispersed in 5 mL of absolute ethanol. The mixture was stirred at room temperature for 12 h. After that, the catalyst of Fe_3_O_4_@O_2_Si[PrMIM]Cl·AlCl_3_ was washed with ethanol (2 × 5 mL) and put into the desiccator overnight. The dark-brown solid obtained was ground into a homogeneous fine powder and stored in the desiccator before using.

### 3.2. General Procedure for the Friedel–Crafts Sulfonylation

The aromatic compound (1.0 mmol), sulfonyl chloride/sulfonic anhydride (1.0 mmol, and Fe_3_O_4_@O_2_Si[PrMIM]Cl·AlCl_3_ (0.2 g) were added into a 5 mL round-bottom flask assembled with the condenser. The reaction mixture was heated at 110 °C for a specific period of time. After cooling down, the mixture of products was extracted with ethyl acetate (4 × 5 mL), and the solid catalyst was collected by using a magnetic bar. The organic phase was rinsed with water (2 × 10 mL) and dried with anhydrous Na_2_SO_4_. After that, the removal of the solvent by rotary evaporation was performed to obtain the crude product. The product was purified by column chromatography using eluent as a mixture of *n*-hexane and ethyl acetate (8:2 *v*/*v*).

### 3.3. Spectroscopic Data

The identification and purity of all products reported were determined by ^1^H-NMR, ^13^C-NMR, and HRMS. The well-known compounds **3a** [63], **3a′** [63], **3b** [64], **3b′** [65], **3c** [64], **3d** [64], **3e** [66], **3e′** [65], **3f** [64], **3f′** [65], **3g** [64], **3g′** [67], **3h** [68], **3i** [69], **3m** [70], **3m′** [71], **3n** [32], **3o** [70], **3s** [30], and **3u** [62] were found to be compatible with the previous literature. The unknown products are described below (Appendix A).

**1-((4-Chlorophenyl)sulfonyl)-2-methylbenzene (3c′)**: White solid; m.p.: 137–138 °C. ^1^H NMR (500 MHz, CDCl_3_) δ (ppm) 8.19 (dd, *J* = 8.0 Hz, *J* = 1.5 Hz, 1H), 7.81–7.78 (m, 2H), 7.51–7.46 (m, 3H), 7.40 (t, *J* = 7.5 Hz, 1H), 7.24 (d, *J* = 7.5 Hz, 1H), 2.44 (s, 3H). ^13^C NMR (125 MHz, CDCl_3_): δ (ppm) 140.0, 139.8, 138.7, 138.1, 134.0, 132.9, 129.6, 129.5, 129.3, 126.8, 20.4. HRMS-ESI: *m*/*z* [M + Na]^+^ calcd. for C_13_H_11_O_2_SCl, 289.0066; found, 289.0101 (Appendix A).

**1-Methyl-2-((4-nitrophenyl)sulfonyl)benzene (3d′)**: White solid; m.p.: 106–108 °C. ^1^H NMR (500 MHz, CDCl_3_): δ (ppm) 8.35–8.33 (m, 2H), 8.25 (dd, *J* = 7.5 Hz, *J* = 1.0 Hz, 1H), 8.05–8.03 (m, 2H), 7.55 (td, *J* = 7.5 Hz, *J* = 1.5 Hz, 1H), 7.46 (t, *J* = 7.5 Hz, 1H), 7.28 (d, *J* = 7.5 Hz, 1H), 2.44 (s, 3H). ^13^C NMR (125 MHz, CDCl_3_): δ (ppm) 150.5, 147.3, 138.4, 137.6, 134.7, 133.2, 130.0, 129.1, 127.1, 124.5, 20.4. HRMS-ESI: *m*/*z* [M + Na]^+^ calcd. for C_13_H_11_NO_4_S, 300.0377; found, 300.0321.

**1-((4-Chlorophenyl)sulfonyl)-2-methoxybenzene (3h′)**: White solid; m.p.: 139–141 °C. ^1^H NMR (500 MHz, CDCl_3_): δ (ppm) 8.14 (dd, *J* = 8.0 Hz, *J* = 1.5 Hz, 1H), 7.90 (d, *J* = 8.5 Hz, 2H), 7.57–7.54 (m, 1H), 7.45 (d, *J* = 8.5 Hz, 2H), 7.11 (t, *J* = 7.5 Hz, 1H), 6.91 (d, *J* = 7.5 Hz, 1H), 3.78 (s, 3H). ^13^C NMR (125 MHz, CDCl_3_): δ (ppm) 157.2, 140.3, 139.7, 135.9, 130.1, 130.0, 128.9, 128.8, 120.8, 112.7, 56.1. HRMS-ESI: *m*/*z* [M + Na]^+^ calcd. for C_13_H_11_O_3_SCl, 305.0015; found, 305.0004.

**1-Methoxy-2-((4-nitrophenyl)sulfonyl)benzene (3i′)**: White solid; m.p.: 164–165 °C. ^1^H NMR (500 MHz, CDCl_3_): δ (ppm) 8.33–8.31 (m, 2H), 8.18–8.14 (m, 3H), 7.61–7.59 (m, 1H), 7.18–7.14 (m, 1H), 6.93 (d, *J* = 8.0 Hz, 1H), 3.78 (s, 3H). ^13^C NMR (125 MHz, CDCl_3_): δ (ppm) 157.3, 147.5, 136.6, 130.3, 129.9, 128.8, 123.9, 121.1, 115.1, 112.8, 56.2. HRMS-ESI: *m*/*z* [M + Na]^+^ calcd. for C_13_H_11_O_5_SN, 316.0256; found, 316.0223.

**1-(Ethylsulfonyl)-4-methoxybenzene (3j)**: White solid; m.p.: 56–58 °C. ^1^H NMR (500 MHz, CDCl_3_): δ (ppm) 7.83 (d, *J* = 9.0 Hz, 2H), 7.02 (d, *J* = 9.0 Hz, 2H), 3.89 (s, 3H), 3.08 (q, *J* = 7.5 Hz, 2H), 1.26 (t, *J* = 7.5 Hz, 3H). ^13^C NMR (125 MHz, CDCl_3_): δ (ppm) 163.9, 130.5, 130.4, 114.6, 55.8, 51.0, 7.7. HRMS-ESI: *m*/*z* [M + H]^+^ calcd. for C_9_H_12_O_3_S, 201.0585; found, 201.0585.

**1-(Ethylsulfonyl)-2-methoxybenzene (3j′)**: White solid; m.p.: 88–90 °C. ^1^H NMR (500 MHz, CDCl_3_): δ (ppm) 7.96 (dd, *J* = 8.0 Hz, *J* = 2.0 Hz, 1H), 7.61–7.57 (m, 1H), 7.12–7.09 (m, 1H), 7.04 (d, *J =* 8.5 Hz, 1H), 3.98 (s, 3H), 3.37 (q, *J* = 7.5 Hz, 2H), 1.24 (t, *J* = 7.5 Hz, 3H). ^13^C NMR (125 MHz, CDCl_3_): δ (ppm) 157.4, 135.5, 130.9, 126.4, 120.8, 112.3, 56.3, 48.7, 7.1. HRMS-ESI: *m*/*z* [M + H]^+^ calcd. for C_9_H_12_O_3_S, 201.0585; found, 201.0583.

**1-(Isobutylsulfonyl)-4-methoxybenzene (3k)**: Light brown liquid. ^1^H NMR (500 MHz, CDCl_3_): δ (ppm) 7.84 (d, *J* = 9.0 Hz, 2H), 7.01 (d, *J* = 9.0 Hz, 2H), 3.88 (s, 3H), 2.96 (d, *J* = 6.5 Hz, 2H), 2.21–2.17 (m, 1H), 1.04 (d, *J* = 6.5 Hz, 6H). ^13^C NMR (125 MHz, CDCl_3_): δ (ppm) 163.8, 132.0, 130.2, 114.6, 64.5, 55.8, 24.3, 22.9. HRMS-ESI: *m*/*z* [M + H]^+^ calcd. for C_11_H_16_O_3_S, 229.0898; found, 229.0896.

**1-(Isobutylsulfonyl)-2-methoxybenzene (3k′)**: Light brown liquid. ^1^H NMR (500 MHz, CDCl_3_): δ (ppm) 7.97 (dd, *J* = 8.0 Hz, *J* = 2.0 Hz, 1H), 7.60–7.56 (m, 1H), 7.10 (td, *J* = 7.5 Hz, *J* = 1.0 Hz, 1H), 7.04 (d, *J* = 8.5 Hz, 1H), 3.98 (s, 3H), 3.25 (d, *J* = 6.5 Hz, 2H), 2.23–2.18 (m, 1H), 1.03 (d, *J* = 6.5 Hz, 6H). ^13^C NMR (125 MHz, CDCl_3_): δ (ppm) 157.3, 135.4, 130.3, 128.1, 120.8, 112.3, 62.3, 56.3, 24.1, 22.7. HRMS-ESI: *m*/*z* [M + H]^+^ calcd. for C_11_H_16_O_3_S, 229.0898; found, 229.0896.

**1-Methoxy-4-(octylsulfonyl)benzene (3l)**: Light brown liquid. ^1^H NMR (500 MHz, CDCl_3_): δ (ppm) 7.84–7.81 (m, 2H), 7.03–7.00 (m, 2H), 3.88 (s, 3H), 3.06–3.03 (m, 2H), 1.70–1.67 (m, 2H), 1.35–1.32 (m, 2H), 1.27–1.23 (m, 8H), 0.86 (t, *J* = 7.0 Hz, 3H). ^13^C NMR (125 MHz, CDCl_3_): δ (ppm) 163.8, 131.1, 130.4, 114.6, 56.8, 55.8, 31.8, 29.1, 29.0, 28.4, 23.0, 22.7, 14.2. HRMS-ESI: *m*/*z* [M + H]^+^ calcd. for C_15_H_24_O_3_S, 285.1524; found, 285.1522.

**1-Methoxy-2-(octylsulfonyl)benzene (3l′)**: Light brown liquid. ^1^H NMR (500 MHz, CDCl_3_): δ (ppm) 7.96 (dd, *J* = 8.0 Hz, *J* = 2.0 Hz, 1H), 7.59–7.57 (m, 1H), 7.10 (td, *J* = 7.5 Hz, *J* = 1.0 Hz, 1H), 7.05 (d, *J* = 8.5 Hz, 1H), 3.98 (s, 3H), 3.35–3.32 (m, 2H), 1.69–1.66 (m, 2H), 1.37–1.33 (m, 2H), 1.28–1.23 (m, 8H), 0.86 (t, *J* = 7.0 Hz, 3H). ^13^C NMR (125 MHz, CDCl_3_): δ (ppm) 157.5, 135.5, 130.8, 129.2, 120.9, 112.4, 56.4, 54.6, 31.8, 29.1, 29.0, 28.4, 22.7, 22.5, 14.2. HRMS-ESI: *m*/*z* [M + H]^+^ calcd. for C_15_H_24_O_3_S, 285.1524; found: 285.1524.

**2,3-Dimethyl-1-tosylbenzene (3o′)**: White solid; m.p.: 130–132 °C. ^1^H NMR (500 MHz, CDCl_3_): δ (ppm) 8.07 (d, *J* = 8.0 Hz, 1H), 7.73 (d, *J* = 8.5 Hz, 2H), 7.37 (d, *J* = 7.5 Hz, 1H), 7.29–7.27 (m, 3H), 2.41 (s, 3H), 2.35 (s, 3H), 2.26 (s, 3H). ^13^C NMR (125 MHz, CDCl_3_): δ (ppm) 143.9, 139.7, 139.5, 139.0, 136.4, 135.2, 129.8, 127.8, 127.5, 125.9, 21.7, 20.5, 16.1. HRMS-ESI: *m*/*z* [M + H]^+^ calcd. for C_15_H_16_O_2_S, 261.0949; found, 261.0954.

**2,4-Dimethoxy-1-tosylbenzene (3p)**: White solid; m.p.: 159–161 °C. ^1^H NMR (500 MHz, CDCl_3_): δ (ppm) 8.04 (d, *J* = 8.5 Hz, 1H), 7.80 (d, *J* = 8.5 Hz, 2H), 7.23 (d, *J* = 8.0 Hz, 2H), 6.55 (dd, *J* = 8.5 Hz, *J* = 2.0 Hz, 1H), 6.36 (d, *J* = 2.0 Hz, 1H), 3.81 (s, 3H), 3.72 (s, 3H), 2.38 (s, 3H). ^13^C NMR (125 MHz, CDCl_3_): δ (ppm) 165.6, 158.7, 143.5, 139.4, 131.7, 129.2, 128.3, 121.9, 104.7, 99.6, 56.0, 55.8, 21.7. HRMS-ESI: *m*/*z* [M + Na]^+^ calcd. for C_15_H_16_O_4_S, 315.0667; found, 315.0632.

**1,3-Dimethoxy-2-tosylbenzene (3p′)**: White solid; m.p.: 104–106 °C. ^1^H NMR (500 MHz, CDCl_3_): δ (ppm) 7.84 (d, *J* = 8.0 Hz, 2H), 7.37 (t, *J* = 8.5 Hz, 1H), 7.24 (d, *J* = 8.0 Hz, 2H), 6.54 (d, *J* = 8.5 Hz, 2H), 3.77 (s, 6H), 2.39 (s, 3H). ^13^C NMR (125 MHz, CDCl_3_): δ (ppm) 159.5, 143.0, 141.7, 134.8, 130.9, 128.8, 127.4, 118.4, 105.4, 56.5, 21.5. HRMS-ESI: *m*/*z* [M + Na]^+^ calcd. for C_15_H_16_O_4_S, 315.0667; found, 315.0642.

**1,4-Dimethoxy-2-tosylbenzene (3q)**: White solid; m.p.: 111–113 °C. ^1^H NMR (500 MHz, CDCl_3_): δ (ppm) 7.85 (d, *J* = 8.5 Hz, 2H), 7.68 (d, *J* = 3.0 Hz, 1H), 7.27 (d, *J* = 8.0 Hz, 2H), 7.06 (dd, *J* = 9.0 Hz, *J* = 3.0 Hz, 1H), 6.84 (d, *J* = 9.0 Hz, 1H), 3.84 (s, 3H), 3.71 (s, 3H), 2.41 (s, 3H). ^13^C NMR (125 MHz, CDCl_3_): δ (ppm) 153.5, 151.4, 144.0, 138.7, 130.1, 129.3, 128.6, 121.7, 114.5, 113.9, 56.7, 56.3, 21.7. HRMS-ESI: *m*/*z* [M + Na]^+^ calcd. for C_15_H_16_O_4_S, 315.0667; found, 315.0700.

**4-((4-Chlorophenyl)sulfonyl)phenol (3r)**: White solid, m.p.: 146–147 °C, ^1^H NMR (500 MHz, CDCl_3_): δ (ppm) 7.83 (d, *J* = 8.5 Hz, 2H), 7.80 (d, *J* = 9.0 Hz, 2H’), 7.45 (d, *J* = 8.5 Hz, 2H), 6.91 (d, *J* = 9.0 Hz, 2H). ^13^C NMR (125 MHz, CDCl_3_): δ (ppm) 160.5, 140.9, 139.8, 132.8, 130.3, 129.7, 128.9, 116.4. HRMS-ESI: *m*/*z* [M + Na]^+^ calcd. for C_12_H_9_O_3_SCl, 290.9859; found, 290.9894.

**2-((4-Chlorophenyl)sulfonyl)phenol (3r′)**: White solid, m.p.: 158–159 °C, ^1^H NMR (500 MHz, CDCl_3_): δ (ppm) 9.11 (s, 1H), 7.88–7.86 (m, 2H), 7.63 (dd, *J* = 8.0 Hz, *J* = 1.5 Hz, 2H′), 7.51–7.45 (m, 3H), 7.01–6.96 (m, 2H). ^13^C NMR (125 MHz, CDCl_3_): δ (ppm) 155.9, 140.5, 140.2, 136.4, 129.8, 129.1, 128.3, 123.2, 121.0, 119.3. HRMS-ESI: *m*/*z* [M + Na]^+^ calcd. for C_12_H_9_O_3_SCl, 290.9859; found, 290.9895.

**2-(Phenylsulfonyl)naphthalene (3s′)**: White solid, m.p.: 123–125 °C, ^1^H NMR (500 MHz, CDCl_3_): δ (ppm) 8.58 (s, 1H), 7.99 (t, *J* = 7.5 Hz, 3H_′_), 7.93 (d, *J* = 9.0 Hz, 1H), 7.88–7.84 (m, 2H), 7.64–7.62 (m, 2H), 7.56–7.50 (m, 3H). ^13^C NMR (125 MHz, CDCl_3_): δ (ppm) 138.3, 134.9, 133.1, 132.2, 129.5, 129.3, 129.2, 129.1, 129.0, 128.7, 127.8, 127.6, 127.5, 122.6. HRMS-ESI: *m*/*z* [M + Na]^+^ calcd. for C_16_H_12_O_2_S, 291.0456; found, 291.0445.

**4-(Phenylsulfonyl)dibenzo[b,d]thiophene (3t′)**: White solid, ^1^H NMR (400 MHz, CDCl_3_): δ (ppm) 8.34 (d, *J* = 7.6 Hz, 1H), 8.19 (d, *J* = 7.6 Hz, 1H′), 8.15 (d, *J* = 7.6 Hz, 1H), 8.10 (d, *J* = 8.0 Hz, 2H), 7.91 (d, *J* = 7.6 Hz, 1H), 7.62 (t, *J* = 7.6 Hz, 1H), 7.53–7.48 (m, 5H). ^13^C NMR (100 MHz, CDCl_3_): δ (ppm) 141.4, 140.8, 138.5, 138.3, 135.7, 134.4, 134.0, 131.3, 129.6, 128.3, 128.1, 126.6, 125.4, 125.3, 123.1, 122.2. MS (C_18_H_12_O_2_S_2_): *m*/*z* = 324[M]^+^ (78%), 199 (65%), 183 (39%), 171 (63%), 139 (100%), 77 (40%), 51 (32%).

**4-Methoxy-2-methyl-1-tosylbenzene (3v)**: White solid; m.p.: 117–119 °C. ^1^H NMR (500 MHz, CDCl_3_): δ (ppm) 8.15 (d, *J* = 9.0 Hz, 1H), 7.71 (d, *J* = 8.0 Hz, 2H), 7.27 (d, *J* = 7.5 Hz, 2H), 6.85 (dd, *J* = 8.5 Hz, *J* = 2.5 Hz, 1H), 6.71 (d, *J* = 2.5 Hz, 1H), 3.83 (s, 3H), 2.39 (s, 6H). ^13^C NMR (125 MHz, CDCl_3_): δ (ppm) 163.5, 143.7, 140.3, 139.3, 132.0, 131.2, 129.7, 127.6, 118.2, 111.1, 55.6, 21.7, 20.6. HRMS-ESI: *m*/*z* [M + Na]^+^ calcd. for C_15_H_16_O_3_S, 299.0718; found, 299.0715.

**2-Methoxy-4-methyl-1-tosylbenzene (3v′)** white solid; m.p.: 128–130 °C. ^1^H NMR (500 MHz, CDCl_3_): δ (ppm) 8.00 (d, *J* = 8.0 Hz, 1H), 7.83 (d, *J* = 8.5 Hz, 2H), 7.25 (d, *J* = 7.5 Hz, 2H), 6.89 (d, *J* = 8.0 Hz, 1H), 6.68 (s, 1H), 3.74 (s, 3H), 2.40 (s, 1H), 2.37 (s, 3H). ^13^C NMR (125 MHz, CDCl_3_): δ (ppm) 156.9, 146.5, 143.4, 138.9, 129.7, 129.0, 128.2, 126.5, 121.1, 113.0, 55.7, 21.8, 21.4. HRMS-ESI: *m*/*z* [M + Na]^+^ calcd. for C_15_H_16_O_3_S, 299.0718; found, 299.0746.

**1-Methoxy-3-methyl-1-tosylbenzene (3v″)**: White solid; m.p.: 107–109 °C. ^1^H NMR (500 MHz, CDCl_3_): δ (ppm) 7.79 (d, *J* = 8.5 Hz, 2H, 7.33 (t, *J* = 8.0 Hz, 1H), 7.25 (d, *J* = 8.0 Hz, 2H), 6.87 (d, *J* = 7.5 Hz, 1H), 6.73 (d, *J* = 8.5 Hz, 1H), 3.61 (s, 3H), 2.84 (s, 3H), 2.40 (s, 3H). ^13^C NMR (125 MHz, CDCl_3_): δ (ppm) 158.3, 143.2, 141.2, 141.1, 133.8, 128.9, 128.3, 127.4, 125.5, 110.9, 56.0, 22.4, 21.6. HRMS-ESI: *m*/*z* [M + Na]^+^ calcd. for C_15_H_16_O_3_S, 299.0718; found, 299.0702.

**1,2-Dimethoxy-4-tosylbenzene (3w)** white solid; m.p.: 130–132 °C. ^1^H NMR (500 MHz, CDCl_3_): δ (ppm) 7.80 (d, *J* = 8.5 Hz, 2H), 7.55 (dd, *J* = 8.5 Hz, *J* = 2.0 Hz, 1H), 7.37 (d, *J* = 2.5 Hz, 1H), 7.28 (d, *J* = 8.0 Hz, 2H), 6.91 (d, *J* = 8.5 Hz, 1H), 3.91 (s, 3H), 3.90 (s, 3H), 2.39 (s, 3H). ^13^C NMR (125 MHz, CDCl_3_): δ (ppm) 152.9, 149.3, 143.8, 139.4, 133.6, 129.8, 127.3, 121.7, 110.9, 109.9, 56.3, 56.2, 21.5. HRMS-ESI: *m*/*z* [M + H]^+^ calcd. for C_15_H_16_O_4_S, 293.0847; found, 293.0850.

**1,2-Dimethoxy-3-tosylbenzene (3w′)**: White solid; m.p.: 128–130 °C. ^1^H NMR (500 MHz, CDCl_3_): δ (ppm) 7.85 (d, *J* = 8.0 Hz, 2H), 7.70 (d, *J* = 8.0 Hz, 1H), 7.26 (d, *J* = 7.5 Hz, 2H), 7.18 (t, *J* = 8.0 Hz, 1H), 7.11 (d, *J* = 8.5 Hz, 1H), 3.86 (s, 3H), 3.83 (s, 3H), 2.39 (s, 3H). ^13^C NMR (125 MHz, CDCl_3_): δ (ppm) 153.8, 147.4, 144.0, 139.1, 135.8, 129.4, 128.3, 123.9, 120.6, 117.8, 61.5, 56.3, 21.7. HRMS-ESI: *m*/*z* [M + H]^+^ calcd. for C_15_H_16_O_4_S, 293.0847; found, 293.0846.

## 4. Conclusions

Using contemporary green chemistry, a chloroaluminate ionic liquid immobilized on magnetic nanoparticles was developed and applied in the solvent-free sulfonylation of substituted aromatic compounds with sulfonyl chlorides, as well as *p*-toluenesulfonic anhydride, to afford sulfones in moderate to good yields. The Friedel–Crafts sulfonylation had preferred arenes and sulfonyl chlorides with electron-donating substituents. The more electron-donating substituents on the aromatic rings, the more the yields of sulfones, and the shorter the reaction times. In addition, another interesting result was that the size of the Fe_3_O_4_ particles, which originally were around 20 nm in diameter, became smaller, in the range of 6–14 nm in diameter, owing to the immobilization of the chloroaluminate ionic liquid on the particles. Furthermore, Fe_3_O_4_@O_2_Si[PrMIM]Cl·AlCl_3_ is an ecofriendly, efficient, and highly recyclable catalyst, especially evidenced by the yields of sulfones without a significant drop after four catalytic cycles of recovery and reuse.

## Data Availability

Not applicable.

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
