# Peer review of "Chloroaluminate Ionic Liquid Immobilized on Magnetic Nanoparticles as a Heterogeneous Lewis Acidic Catalyst for the Friedel–Crafts Sulfonylation of Aromatic Compounds"

_molecules, 2022, doi:10.3390/molecules27051644_

Round 1

Reviewer 1 Report

The MS molecules-1593700 entitled “Chloroaluminate ionic liquid immobilized on magnetic nano-particle as a heterogeneous Lewis acidic catalyst for the Friedel−Crafts sulfonylation of aromatic compounds” written by Ngoc-Lan Thi Nguyen, Quoc-Anh Nguyen, Tien Khoa Le, Thi Xuan Thi Luu, Kim-Ngan Thi Tran, and Phuoc-Bao Pham focuses on the preparation of Fe3O4@O2Si[PrMIM]Cl.AlCl3, its characterization with XRD, FT-IR spectroscopy, SEM, TEM, EDX, TGA and VSM, and application as catalyst in the Friedel-Crafts sulfonylation of various aromatic compounds with either sulfonyl chlorides or p-toluenesulfonic anhydride. Extended supporting information containing HRMS and NMR spectra of the reaction products belongs to the MS as well. The MS is interesting. However, a few questions exist that should be answered before publication in molecules.
As shown in Scheme 2 and given in the materials and methods part, ammonia was used for preparation of Fe3O4@O2Si[PrMIM]Cl.  However, the base ammonia may react with imidazolium. It would be interesting to get information to which extent the reaction of ammonia with the imidazolium structure occurred under the conditions used for the preparation of Fe3O4@O2Si[PrMIM]Cl.
The numbering (3a), (3b), (3c), and (3d) is not clearly shown in the single pictures of Figure 3.  Authors should make more clear which number corresponds to the single picture in Figure 3.
The TGA curve of Fe3O4@O2Si[PrMIM]Cl.AlCl3 depicted in Figure 5 shows remaining water content in this catalyst. Remaining water may influence the Friedel-Crafts sulfonylation of the aromatic compounds because it may react with starting material such as sulfonyl chlorides and p-toluenesulfonic anhydride.  Discussion regarding the influence of remaining water in the catalyst is missing in the MS.
The authors wrote “The FT-IR analysis demonstrated that the functional groups of recovered catalyst at the fourth recycled time was compatible with that of the fresh Fe3O4@O2Si[PrMIM]Cl.AlCl3  (Figure 7)” on page 11, line 228 – 229.  As the FT-IR spectra differ in selected wavenumber regions, a more detailed information would be helpful for the reader regarding both the functional groups including the corresponding wavenumbers on the one hand and discussion of differences on the other hand. 
Furthermore, authors wrote “All commercially available chemicals used were from Acros, Sigma-Aldrich …” on page 12, line 246. This information is too general. It would be helpful to the reader to specify both the chemicals and the companies from which the single chemicals were purchased.
Moreover, the conclusions read like a summary.  Therefore, improvement of the conclusions is necessary.

Author Response

Referee No 1.

We would like to answer to Referee as follows:

- Role of aq. ammonia as promoter for hydroxy group on the surface of Fe3O4 particles via hydrogen bonding in order to attack to silane center of functionalized ionic liquid. It is similar with the previous literature.[56]

[56] Safari, J.; Zarnegar, Z., Immobilized ionic liquid on superparamagnetic nanoparticles as an effective catalyst for the synthesis of tetrasubstituted imidazoles under solvent-free conditions and microwave irradiation. C. R. Chim. 2013, 16 (10), 920-928.  

- The numbering (3a), (3b), (3c), and (3d) is not clearly shown in the single pictures of Figure 3.  Authors should make more clear which number corresponds to the single picture in Figure 3.

Modified in the manuscript (Figure 3).

- The TGA curve of Fe3O4@O2Si[PrMIM]Cl.AlCl3 depicted in Figure 5 shows remaining water content in this catalyst. Remaining water may influence the Friedel-Crafts sulfonylation of the aromatic compounds because it may react with starting material such as sulfonyl chlorides and p-toluenesulfonic anhydride.  Discussion regarding the influence of remaining water in the catalyst is missing in the MS.

+ Loss of weight approx. 7% at 100 oC in TGA could be come from the storage of complete catalyst during TGA analysis (10-14 days). The percentage of water in the composition of complete catalyst is not much as that if it is used immediately as soon as its preparation. Its efficiency was illustrated via the obtained yields in Tables. The presence of water in complete catalyst were found in the previous literature as follows:

[1] Simin Nazari, Shervin Saadat, Pegah Kazemian Fard, Maryam Gorjizadeh, Eshagh Rezaee Nezhad,

Mozhgan Afshari. Monatsh Chem. 2013,144:1877–1882.

[2] Hai Truong Nguyen, Ngoc-Phuong Thi Le, Duy-Khiem Nguyen Chau, Phuong Hoang Tran. RSC Adv., 2018, 8, 35681–35688.

- The authors wrote “The FT-IR analysis demonstrated that the functional groups of recovered catalyst at the fourth recycled time was compatible with that of the fresh Fe3O4@O2Si[PrMIM]Cl.AlCl3  (Figure 7)” on page 11, line 228 – 229.  As the FT-IR spectra differ in selected wavenumber regions, a more detailed information would be helpful for the reader regarding both the functional groups including the corresponding wavenumbers on the one hand and discussion of differences on the other hand. 

+ Modified in the manuscript (Figure 7).

- Furthermore, authors wrote “All commercially available chemicals used were from Acros, Sigma-Aldrich …” on page 12, line 246.

+ Modified in the manuscript as follows:

“Sulfonyl chlorides (benzenesulfonyl chloride, 4-methylbenzenesulfonyl chloride, ethanesulfonyl chloride, isobutanesulfonyl chloride…), anhydrous aluminum chloride, arenes (anisole, 1,3-dimethoxybenzene, naphthalene, chlorobenzene,…), (3-chloropropyl)trimethoxysilane and 1-methylimidazole were from Sigma-Aldrich, besides p-toluenesulfonic anhydride and isomer of xylene were from Acros. All commercially available chemicals were analyzed for authenticity and purity by GC/MS before being used.”

- The conclusions read like a summary.  Therefore, improvement of the conclusions is necessary.

+ Modified in the manuscript as follows:

“With contemporary green chemistry, chloroaluminate ionic liquid immobilized on magnetic nanoparticles have been developed and applied for the solvent-free sulfonylation of substituted aromatic compounds with sulfonyl chlorides as well as p-toluenesulfonic anhydride to afford sulfones in moderate to good yields. The Friedel-Crafts sulfonylation has preferred arenes as well as sulfonyl chlorides with electronic donating substituents. The more electronic donating substituents on aromatic rings, the more yields of sulfones and the shorter reaction times. In addition, another interesting thing is that the size of Fe3O4 particles around 20 nm diameter became smaller in the range of 6-14 nm diameter owing to the immobilization of chloroaluminate ionic liquid on particles. Furthermore, Fe3O4@O2Si[PrMIM]Cl×AlCl3 is eco-friendly, efficient and highly recyclable catalyst, especially the yields of sulfones without drop significantly after four catalytic cycles of recovery and reuse.”

Reviewer 2 Report

In this manuscript the authors report the synthesis of aromatic sulfones using benzenesulfonyl chloride or sulfonic anhydride. As a catalyst for this Friedel-Crafts type reaction, an ionic liquid supported on magnetic nanoparticles is used. In this way the catalysis becomes heterogeneous and the catalyst can be easily recovered from the reaction environment.

In my opinion the authors show an accurate characterization of the catalyst using many different techniques (XRD, FT-IR, SEM, TGA, VSM), which demonstrate its correct structure. They test this catalyst in the Fridel-Crafts sulfonylation reaction between aromatic compuond and benzenesulfonyl chloride or sulfinc anhydride. The scope is really very wide and they get good results. Finally, they demonstrate how this catalyst can be used for at least 4 cycles without losing activity. For these reasons I believe that this manuscript can be accepted without further revision.

Author Response

Thank you for your nice comments

Reviewer 3 Report

The authors report interesting and correct study and I recommend to publish it.

I just have one comment regarding the terminology "ionic liquid". Normally ionic liquid is a salt, which exists in liquid form at ambient conditions (or has low melting temperature). The author immobilize 3-methyl-1-propyl-1H-imidazole-3-ium chloride via propyl bridge – indeed, this compound can be an ionic liquid in pure state, however in no case it forms a liquid on the surface of Fe3O4 nanoparticles. With the same success immobilization of, for example, n-hexyl can be called as immobilization of "hydrophobic liquid" because hexane is liquid, and formation of –OH groups on the surface can be called as immobilization of hydrophilic liquid, because H2O is liquid. The fact that the compound is well-known as ionic liquid does not give grounds to call the fragment of this compound (which is not liquid) as ionic liquid.

Author Response

- I just have one comment regarding the terminology "ionic liquid". Normally ionic liquid is a salt, which exists in liquid form at ambient conditions (or has low melting temperature). The author immobilize 3-methyl-1-propyl-1H-imidazole-3-ium chloride via propyl bridge – indeed, this compound can be an ionic liquid in pure state, however in no case it forms a liquid on the surface of Fe3O4 nanoparticles. With the same success immobilization of, for example, n-hexyl can be called as immobilization of "hydrophobic liquid" because hexane is liquid, and formation of –OH groups on the surface can be called as immobilization of hydrophilic liquid, because H2O is liquid. The fact that the compound is well-known as ionic liquid does not give grounds to call the fragment of this compound (which is not liquid) as ionic liquid.

+The appearance of complete catalyst has been modified and described in the manuscript as follows:

“The dark-brown solid obtained was ground into homogeneous fine powder and stored in the desiccator before using.”